# The Increasing Prognostic and Predictive Roles of the Tumor Primary Chemosensitivity Assessed by CA-125 Elimination Rate Constant K (KELIM) in Ovarian Cancer: A Narrative Review

**DOI:** 10.3390/cancers14010098

**Published:** 2021-12-25

**Authors:** Ambroise Lauby, Olivier Colomban, Pauline Corbaux, Julien Peron, Lilian Van Wagensveld, Witold Gertych, Naoual Bakrin, Pierre Descargues, Jonathan Lopez, Vahan Kepenekian, Olivier Glehen, Charles Andre Philip, Mojgan Devouassoux-Shisheboran, Michel Tod, Gilles Freyer, Benoit You

**Affiliations:** 1Oncology Department, CITOHL, Lyon-Sud Hospital, Cancer Institute of Hospices Civils de Lyon (IC-HCL), Hospices Civils de Lyon, 69495 Lyon, France; ambroise.lauby@chu.lyon.fr (A.L.); pauline.corbaux@chu-lyon.fr (P.C.); julien.peron@chu-lyon.fr (J.P.); gilles.freyer@chu-lyon.fr (G.F.); 2Lyon-Sud Medicine School, University of Lyon, University Claude Bernard Lyon 1, 69008 Lyon, France; olivier.colomban@chu-lyon.fr (O.C.); naoual.bakrin@chu-lyon.fr (N.B.); vahan.kepenekian@chu-lyon.fr (V.K.); olivier.glehen@chu-lyon.fr (O.G.); michel.tod@chu-lyon.fr (M.T.); 3Department of Research, Netherlands Comprehensive Cancer Organization (IKNL), 3511 DT Utrecht, The Netherlands; l.vanwagensveld@iknl.nl; 4GROW, School for Oncology and Developmental Biology, Maastricht University, 6200 MD Maastricht, The Netherlands; 5Department of Gynecological Surgery, Lyon-Sud Hospital, Hospices Civils de Lyon, 69495 Lyon, France; witold.gertych@chu-lyon.fr (W.G.); pierre.descargues@chu-lyon.fr (P.D.); 6Departement of Surgical Oncology, Lyon-Sud Hospital, Hospices Civils de Lyon, 69495 Lyon, France; 7Biochemistry and Molecular Biology Department, Lyon-Sud Hospital, Hospices Civils de Lyon, 69495 Lyon, France; jonathan.lopez@chu-lyon.fr; 8Croix Rousse Hospital, Hospices Civils de Lyon, 69004 Lyon, France; charles-andre.philip01@chu-lyon.fr; 9Department of Pathology, Lyon-Sud Hospital, Hospices Civils de Lyon, 69495 Lyon, France; Mojgan.devouassoux@chu-lyon.fr; 10Hospices Civils de Lyon, Pharmacie, Croix Rousse Hospital, Hospices Civils de Lyon, 69004 Lyon, France

**Keywords:** ovarian cancer, KELIM, CA-125, primary chemosensitivity, treatment success

## Abstract

**Simple Summary:**

In patients with advanced ovarian cancers, the standard first-line treatment includes debulking surgery and platinum-based chemotherapy, followed by a maintenance treatment. Contrary to the completeness of the debulking surgery, the prognostic impact of the tumor chemosensitivity in the success of the first-line treatment has been insufficiently addressed due to the lack of a reliable indicator of the primary chemosensitivity, as acknowledged by European consensus conferences. The objective of this narrative review is to present an overview of the modeled CA-125 ELIMination rate constant K (KELIM) calculation based on the longitudinal CA-125 kinetics during the first chemotherapy cycles and its utility as an early marker of tumor primary chemosensitivity. Easily calculable online, KELIM was shown to be a consistent and reproducible early prognostic marker that could be useful for understanding the prognosis of patients and adjusting the medical–surgical treatments.

**Abstract:**

Ovarian cancer is the gynecological cancer with the worst prognosis and the highest mortality rate because 75% of patients are diagnosed with advanced stage III–IV disease. About 50% of patients are now treated with neoadjuvant chemotherapy followed by interval debulking surgery (IDS). In that context, there is a need for accurate predictors of tumor primary chemosensitivity, as it may impact the feasibility of subsequent IDS. Across seven studies with more than 12,000 patients, including six large randomized clinical trials and a national cancer registry, along with a mega-analysis database with 5842 patients, the modeled CA-125 ELIMination rate constant K (KELIM), the calculation of which is based on the longitudinal kinetics during the first three cycles of platinum-based chemotherapy, was shown to be a reproducible indicator of tumor intrinsic chemosensitivity. Indeed, KELIM is strongly associated with the likelihood of complete IDS, subsequent platinum-free interval, progression-free survival, and overall survival, along with the efficacy of maintenance treatment with bevacizumab or veliparib. As a consequence, KELIM might be used to guide more subtly the medical and surgical treatments in a first-line setting. Moreover, it could be used to identify the patients with poorly chemosensitive disease, who will be the best candidates for innovative treatments meant to reverse the chemoresistance, such as cell cycle inhibitors or immunotherapy.

## 1. Introduction

With 184,799 deaths worldwide in 2018, ovarian cancer is the leading cause of death among all gynecological cancers in developed countries [1]. Due to an initially asymptomatic disease, approximately 75% of patients at the time of diagnosis present with advanced tumors (FIGO stages III to IV) known to have a poor prognosis [2]. Despite standard treatment associating debulking surgery and platinum-based chemotherapy, recurrent disease develops in more than 95% of patients within 5 years [3]. Relapse is typically incurable, with most patients receiving multiple additional lines of treatment before ultimately dying from the disease. On the other hand, despite a poor prognosis of advanced-stage disease, it is not uncommon to observe patients surviving more than 10 years [4]. In that context, there is a need for accurate predictors of outcomes in patients to guide treatment decisions in the first-line setting, as acknowledged by ESMO and ESGO consensus conferences [5].

In developed countries, more than 90% of all ovarian cancers have an epithelial origin, designated epithelial ovarian cancer (EOC) [6]. CA-125 antigen is a serum marker that is overexpressed in EOC cells and is found elevated in the blood of 80% to 90% of patients with advanced epithelial ovarian carcinomas. The early longitudinal kinetics of CA-125, known to be related to tumor burden response, has been extensively studied as a prognostic marker of the treatment efficacy [7,8,9,10,11,12]. 

The Gynecologic Cancer InterGroup (GCIG) defined the CA-125 response as a 50% reduction in CA-125 levels maintained for at least 28 days, in patients treated for recurrent disease [13]. However, the relevance of the GCIC criteria was recently questioned in the analyses of the CALYPSO trial in the recurrent setting and of ICON-8 trial in a first-line setting [10,14].

In that context, recent studies have suggested that longitudinal assessment of CA-125 values using mathematical algorithms would be more accurate than single or two threshold-based rules for analyzing CA-125 kinetics [15,16,17].

Based on a semi-mechanistic model and population kinetic approach, the modeled CA-125 ELIMination rate constant K (KELIM) was developed for that purpose. KELIM is a modeled kinetic parameter and is easily accessible to the clinician with an online calculator using at least three values of CA-125 during systemic treatment.

The aim of this narrative review is to present an overview of KELIM utility across the different data published or presented in congress.

## 2. Limitations of the GCIG CA-125 Response Definition

As peritoneal involvement is frequently difficult to assess by conventional imaging examinations, tumor response cannot be frequently assessed using Response Evaluation Criteria in Solid Tumors (RECIST), and CA-125 is therefore a valuable additional tool for early assessment of tumor response in clinical trials.

In 1996, Rustin et al. proposed three accurate definitions of CA-125 response to relate the decrease in the levels of CA-125 to the tumor response in patients receiving initial chemotherapy for ovarian cancer [7]. The definitions were based on 50% or 75% decreases in several samples of CA-125 maintained over at least 28 days. Reliable relationships between radiological partial response as per Gynecologic Oncology Group criteria and CA-125 responses were found.

In 2004, a simplified version of the original formula proposed by Rustin et al. was developed and adopted by the GCIG. It was defined as a CA-125 decline by at least 50% from a pretreatment sample and maintained for at least 28 days [18]. This is still the official definition by GCIG, and it is recommended for the evaluation of the response to systemic treatments in recurrent ovarian cancers in clinical trials [13]. 

However, Lee et al. were the first team to question the relevance of the GCIG CA-125 response criterion on the data of the phase III CALYPSO trial, which compared a combination of pegylated liposomal doxorubicin with carboplatin (CPLD) with standard carboplatin and paclitaxel (CP) in patients with platinum-sensitive recurrent ovarian cancers (ROC) [19]. Progression-free survival was statistically superior in the CPLD arm (hazard ratio 0.821, 95% CI 0.72–0.94, *p* = 0. 005). However, the early CA-125 decline as per the GCIG definition was not associated with the difference in PFS between CPLD and CP [10]. Indeed, early decline was significantly more frequent in patients treated with CP (233 (51.2%)) compared with those treated with CPLD (161 (37.4%)) (odds ratio 1.76, 95% CI 1.35–2.30, *p* < 0.001). These results were contradictory to those expected.

More recently, Morgan et al. reported the lack of accuracy of the GCIG criterion to identify the patients who were likely to benefit from interval debulking surgery (IDS) after neoadjuvant chemotherapy for EOC in the first-line setting of the phase III ICON-8 trial [14,20]. For example, a complete cytoreduction was achieved in 30 of 101 women (30%) without a GCIG CA-125 response and in 290 of 576 women (50%) who experienced a GCIG CA-125 response. Therefore, patients should not be refused IDS based on the lack of GCIG CA-125 response. 

## 3. The Emergence of KELIM, A Novel Modeled Longitudinal CA-125 Kinetic Parameter in Patients with ROC

### 3.1. Semi-Mechanistic Model Building

In 2013, You et al. developed a semi-mechanistic model of the CA-125 kinetics based on a population approach and pharmacokinetic–pharmacodynamic (PK-PD) principles to assess the prognostic value of the longitudinal CA-125 kinetics on the data from the CALYPSO trial [21].

The semi-mechanistic model structure relies on a central compartment where the blood serum concentration of CA-125 is described through a production rate constant KPROD, balanced with an elimination rate constant KELIM. The KPROD is regulated by the effects of the chemotherapy, described with a 2-virtual-compartment model (C1 a central compartment receiving chemotherapy dosing and C2 a transit compartment) on the cancer cells through an indirect effect, characterized by an E50 inhibition rate constant. Because the concentrations of the chemotherapy were not available, a kinetic–pharmacodynamic (K-PD) model was developed to assess the longitudinal kinetics of CA-125 with the administration dose of chemotherapy arbitrarily set to 1, as already carried out for PK models [22].

To normalize their distribution, CA-125 titers were Box–Cox transformed. At least two CA-125 values during the first 50 days of treatment were required. This time frame was arbitrarily selected such that early predictive factors of efficacy were identified.

The model is presented in Figure 1:

This semi-mechanistic model implies that the blood concentration of CA-125 results from a production by the tumor, according to a production rate constant KPROD, and an elimination according to the elimination constant rate KELIM. In the case of systemic treatment, KPROD is inhibited by an indirect effects model characterized by the 50% inhibition effect E50. 

Based on the model described in Figure 1, mathematical equations were built and used to fit the observed kinetics of CA-125 during treatment. In a second time, the following kinetic parameters were estimated on an individual basis: K, the treatment kinetic rate constant (days^−1^); KPROD, the CA-125 tumor production rate constant (IU·mL^−1^·days^−1^); BETA, the tumor growth rate constant (days^−1^); E50, the concentration producing 50% of the maximum effect (IU); and KELIM, the CA-125 elimination rate constant (days^−1^). 

KELIM can be understood as the rate of CA-125 decline during systemic treatment, a kind of “clearance” that would not be related to liver of renal function and would be a reflection of the primary chemosensitivity. The higher KELIM, the faster the CA-125 elimination for a same dose of chemotherapy, and the higher the chemotherapy efficacy.

### 3.2. Application to CALYPSO Dataset

The data from the 895 patients enrolled in this trial were analyzed with the above model [21]. 

After validation of the most accurate model for characterizing longitudinal CA-125 kinetics and confirmation that kinetic parameters were well estimated, the prognostic value of the estimated modeled kinetic parameters was assessed. 

No relationship was found between modeled kinetic parameters and the radiological response rate evaluated according to the RECIST criteria. However, only 25% of patients had assessable lesions using RECIST criteria, thereby limiting the relevance of this outcome.

The data from 875 patients were available for the survival analysis. Three kinetic parameters categorized by medians (< or ≥ median) had predictive value regarding PFS using univariate analyses: the treatment kinetic rate constant K; the CA-125 tumor production rate constant KPROD; and the CA-125 elimination rate constant KELIM (all *p* < 0.001). Using Cox multivariate analysis, only KELIM was significantly associated with PFS (HR 0.53, 95% CI 0.45–0.61, *p* < 0.001).

## 4. KELIM: An Early Marker of the Tumor Primary Chemosensitivity in Ovarian Cancer in First-Line Setting

Following these promising data suggesting that KELIM might be a significant independent prognostic factor for PFS in patients with platinum-sensitive ROC, further assessment of this prognostic marker in the first-line setting was warranted. 

Colomban et al. first reported the prognostic value of KELIM during first-line treatment using the data from three large randomized clinical trials [23]. The data from 2868 patients enrolled in AGO-OVAR 7, AGO-OVAR 9, and ICON-7 trials, which were meant to compare the standard first-line carboplatin paclitaxel regimen (CP) with experimental triplet regimens (CP + gemcitabine in AGO-OVAR 9; topotecan following CP in AGO-OVAR 7; and CP + bevacizumab in ICON 7) were retrospectively assessed. Of note, the chemotherapy was given almost exclusively in an adjuvant setting after initial debulking surgery (only 2% of patients were not operated in ICON 7) [24,25,26].

A simplified model derived from the CALYPSO trial was used to estimate the CA-125 KELIM parameter in these datasets. The time window for assessment of KELIM was extended to 100 days after the start of the chemotherapy to maximize the number of assessable patients with the minimum three available values of CA-125, while remaining a reasonable time period for identifying an early prognostic factor. Moreover, CA-125 titers were log-transformed to normalize their distribution. 

Because the initial model was built in patients with platinum-sensitive ROC, it was necessary to adjust the parameters to patients treated in a first-line setting. The AGO-OVAR 9 database was therefore used as a learning dataset for adjusting the model parameters and for assessing the model accuracy by estimating the standard errors of estimated parameters and performing goodness-of-fit plots. At a second time, the AGO-OVAR 7 and ICON-7 databases were used as external validation sets.

Consistent with CALYPSO trial analysis, KELIM exhibited higher prognostic value of PFS and OS gain than those of the GCIG response criterion. Using univariate analyses, the predictive value index of KELIM for OS and PFS was consistently higher than with GCIG response criterion: PFS, C-index for KELIM = 0.60 (95% CI, 0.58–0.62) vs. 0.49 for the GCIG response criterion (95% CI, 0.48–0.52); OS, C-index = 0.61 (95% CI, 0.59–0.62) vs. 0.51 for the GCIG response criterion (95% CI, 0.50–0.52). With values close to 0.5, the C-index outcomes of the GCIC response criterion was neither discriminatory for OS nor PFS. These data were consolidated in the multivariate C-index analyses since the addition of KELIM to the multivariate model improved the relative apparent performance by 9.7% for PFS and 8.2% for OS, while GCIG response criteria did not provide any benefit.

KELIM exhibited reproducible prognostic information for PFS and OS across the three trials when considered as a continuous or a categorized covariate. To facilitate the interpretation, KELIM was categorized as favorable, intermediate, and unfavorable, according to the values of the terciles found in AGO-OVAR 9 to show the gradual impact of KELIM. Using Kaplan–Meier analyses, median PFS and OS were significantly better in patients with upper KELIM terciles compared with those with lower KELIM terciles in the three trials (Figure 2). Multivariate Cox analyses confirmed the independent and strong prognostic value of KELIM regarding PFS and OS when assessed with the other prognostic factors. KELIM upper tercile was consistently and significantly associated with higher PFS and OS (e.g., for OS in AGO OVAR 9, HR 0.45, 95% CI 0.37–0.55).

Following this publication, other datasets were analyzed to confirm the predictive and prognostic value of KELIM regarding PFS and OS in a first-line setting. 

KELIM was assessed in the CHIVA randomized phase II trial that investigated the addition of nintedanib (a receptor tyrosine kinase inhibitor with potential antiangiogenic and antineoplastic activities) to carboplatin–paclitaxel as neoadjuvant chemotherapy (NACT) for advanced EOC [27,28]. The data from 134 patients were available. KELIM was standardized by the cutoff and able to maximize the prediction of complete IDS likelihood defined by the Youden index. Standardization was a way of normalizing KELIM and providing an easy reading of patient KELIM outcome. Standardized (std) KELIM ≥ 1.0 was considered as favorable; std KELIM (0.5–1) as intermediate; and std KELIM < 0.5 as unfavorable. Using log rank tests, the patients with favorable std KELIM tercile had statistically higher median OS and PFS than patients with less favorable KELIM (e.g., the median OS was 20.4 months, 11.4 months, and 8.4 months in those with favorable, intermediate and unfavorable std KELIM respectively, *p* < 0.01). Multivariate Cox regression model analyses confirmed the independent prognostic value of KELIM with respect to the other prognostic factors. 

In a recent study involving 1582 patients from the Netherlands Cancer Registry and treated with NACT, favorable KELIM was significantly and favorably associated with OS (HR 0.75, 95% CI 0.65–0.87) and PFS (HR 0.77, 95% CI 0.67–0.89) [29].

Moreover, the results of a retrospective analysis of the prognostic value of KELIM regarding the benefit from the fractionated dose-dense chemotherapy in the ICON-8 trial were presented at ASCO 2021 [20,30]. In this trial, patients with EOC were randomly assigned to group 1 (standard carboplatin AUC5-6 and paclitaxel 175 mg/m^2^ every 3 weeks), group 2 (carboplatin as in group 1 and weekly paclitaxel 80 mg/m^2^), or group 3 (weekly carboplatin AUC 2 and paclitaxel 80 mg/m^2^) in a first-line setting. The tumor primary chemosensitivity (by KELIM), along with completeness of debulking surgery, was found to be an independent and prognostic factor of PFS and OS. 

In a pooled analysis of datasets presented above, the prognostic value of KELIM regarding the probability of long-term complete remission (LCR) was assessed after first-line treatment [3]. LCR was defined as complete remission >5 years after initial treatment. The prognostic value of KELIM was assessed in an adjuvant dataset (composed of three phase III trials: AGO OVAR 7, AGO OVAR 9, ICON 7) and a neoadjuvant dataset (Netherlands Cancer Registry). Using multivariate tests, disease stage, completeness of debulking surgery, and std KELIM were all associated with the likelihood of LCR.

The analysis of the GCIG meta-analysis dataset in 5842 patients enrolled in eight randomized phase III trials was recently presented by Corbaux et al. at the 2021 ESMO annual meeting. The strong prognostic and independent prognostic value of KELIM (favorable vs. unfavorable) regarding PFS (HR 0.50, 95% CI 0.44–0.57) and OS (HR 0.45, 95% CI 0.37–0.54) was confirmed [31].

In all studies, the fact that KELIM is strongly associated with PFS and OS suggests that KELIM is an indicator of tumor sensitivity, not only of the platinum-based chemotherapy regimens but also of the subsequent chemotherapy lines. 

In several studies, further analyses were performed to delineate the utility of this marker and some potential applications for routine.

## 5. Potential Utility of KELIM

### 5.1. Potential Utility for Predicting the Likelihood of Complete Interval Debulking Surgery 

The patients with stage III or IV ovarian carcinomas who are not considered to be operable with complete primary debulking surgery are recommended to be treated with a neoadjuvant platinum-based chemotherapy for three or four cycles before planning interval debulking surgery (IDS) [5,32]. NACT increases significantly the rate of complete surgery with no residual disease compared with primary debulking surgery, but its benefits on overall survival and PFS have not yet been established [33,34,35,36]. Obtaining a complete cytoreduction without microscopic residues (CC0 surgery) is already widely recognized as the main goal of the surgery, as it was shown to be a major prognostic factor [37,38,39,40]. As a consequence, the success of the first-line of treatment seems to logically depend on the tumor primary sensitivity to chemotherapy and the likelihood of complete IDS [41].

You et al. investigated the role of the chemosensitivity, assessed with KELIM, relative to the success of first-line medical–surgical treatment using data from the CHIVA trial [27]. Median std KELIM was significantly higher in patients who were operated with complete IDS (1.04) compared with patients operated with incomplete IDS (1.04 vs. 0.54, *p* < 0.01) (Figure 3a). In the final multivariate logistic regression model integrating (1) std KELIM as a continuous covariate; (2) FIGO stage; and (3) radiological response rate at the end of neoadjuvant chemotherapy, only std KELIM was found significantly associated with the likelihood of complete IDS (OR 16.13, 95% CI 5.51-53.38, *p* < 0.001). Based on std KELIM value, the probability of complete IDS can be estimated using a logistic regression curve (Figure 3b).

Consistent with these data, the analysis of the Netherlands Cancer Registry found strong relationships between KELIM and probability of complete surgery, as did the post hoc analysis of VELIA [29,42]. 

We assume that KELIM calculated during the first three to four cycles of neoadjuvant chemotherapy could be used as an indicator of the tumor primary chemosensivitivity to assess the likelihood of subsequent complete interval debulking surgery, especially when the resectability after three cycles of chemotherapy is uncertain or the risk of morbidity is high.

### 5.2. Potential Utility for Medical–Surgical Treatment Adjustment in First-Line Setting 

Despite initial responses to first-line platinum-based chemotherapy in about 70% of patients, recurrent disease occurs in more than 95% of patients [43,44]. Relapse is a detrimental event in the history of the disease, as patients will receive several additional lines of treatment before eventually dying from the disease. 

The platinum-free interval, defined as the time duration between the last cycle of platinum-based chemotherapy and relapse, was shown to be a major prognostic factor for patient survival [45,46]. In the traditional definition, when this interval is >6 months, the disease is considered to be platinum-sensitive, and patients are treated with platinum-based regimens. When this interval is <6 months, the disease is considered to be platinum-resistant, with poor prognosis, and patients are treated with other regimens without platinum. 

There are no validated predictors of the risk of subsequent platinum-resistant relapse after first-line treatment, although it might be a useful parameter for adjusting treatment. Two studies showed that KELIM was significantly associated with the platinum-free interval and the risk of subsequent platinum-resistant relapse.

First, the analysis of the CHIVA trial dataset demonstrated close relationships between std KELIM and the risk of further platinum-resistant relapse [27]. In multivariate logistic regression models, only std KELIM (considered as a continuous covariate) and complete IDS (no vs. yes) were independent prognostic factors regarding the probability of subsequent platinum-resistant relapse (OR 0.13, 95% CI 0.03–0.49 and OR 0.30, 95% CI 0.11–0.76, respectively). Patients with less chemosensitive disease experienced the highest benefit from complete IDS regarding the risk of subsequent platinum-resistant relapse. Conversely, the benefit of complete IDS was minimal in patients with highly chemosensitive disease (Figure 4).

A second study involving 2868 patients from the pooled analysis of ICON7 and AGO-OVAR 7 and 9 was also in agreement with these data [47]. Using multivariate logistic models, continuous KELIM was significantly associated with the risk of subsequent platinum-resistant relapse risk (OR 0.17, 95% CI 0.11–0.25). The impact of disease stage was maximum in patients with unfavorable KELIM and gradually decreased with increasing KELIM, thereby suggesting that the prognostic was mainly driven by the tumor primary chemosensitivity in patients with favorable KELIM. 

On the other hand, the post hoc analysis of ICON 8 suggested that both the tumor primary chemosensitivity and the completeness of debulking surgery were major and complementary prognostic factors. Three subgroups of patients could be identified: (1) a subgroup of patients treated with complete surgery and favorable KELIM, who had the best PFS and OS; (2) a subgroup of patients treated with incomplete surgery and favorable KELIM, or complete surgery and unfavorable KELIM who had intermediate prognosis; (3) a subgroup of patients treated with incomplete surgery and unfavorable KELIM, who had the worse prognosis. The latter subgroup derived a significant benefit from weekly dose-dense chemotherapy [30]. However, these data from a subgroup analysis of a negative trial remain primarily exploratory. 

All together, these outcomes converge for suggesting that KELIM could be useful for adjusting more subtly the first-line treatment, with personalization of management according to the primary chemosensitivity. Indeed, a better understanding of the relative contributions of surgery and medical therapy relative to the overall success might help identify the potential predominant driver of patient prognosis. For example, decision making regarding IDS after neo-adjuvant chemotherapy could be differentially considered according to tumor primary chemosensitivity. Indeed, the relevance of IDS could be questioned in highly chemosensitive patients, especially when the completeness of the planned IDS procedure is uncertain or the risk of morbidity/sequelae related to the surgical procedure is expected to be high. Conversely, obtaining complete IDS appears to be of high importance in patients with poorly chemosensitive disease to improve patient prognosis.

Furthermore, by providing early information on the primary chemosensitivity, KELIM may help better select patients with poorly chemosensitive diseases who are more likely to benefit from therapeutic intensification in future innovative trials.

### 5.3. Potential Utility for Decision-Making Regarding the Maintenance Treatment in First-Line Setting

KELIM may also be interesting for selecting the appropriate maintenance therapy between bevacizumab and PARP inhibitor. Based on the data from two phase III trials, GOG-0218 and ICON-7, bevacizumab was approved in combination with carboplatin and paclitaxel, followed by maintenance as a single-agent for 15 months in patients with stage III–IV ovarian carcinomas [26,48]. Despite this approval, the best patient candidates for bevacizumab prescription remain debated because the PFS benefit was limited, and no OS benefit was found in the intent-to-treat population. In ICON 7, Oza et al. identified a subpopulation of patients with high-risk disease (stage IV, or those with unoperated or suboptimally debulked (>1 cm) stage III disease) who had OS benefit with the addition of bevacizumab (39.3 vs. 34.5 months *p* = 0.03). In GOG-0218, an OS benefit related to bevacizumab addition was identified in patients with stage IV disease only. 

Based on data from 1386 patients from the ICON 7 trial, Colomban et al. assessed the prognostic value of KELIM regarding OS benefit with bevacizumab [49]. The OS of patients within high- and low-risk disease groups was assessed according to treatment arms and std KELIM (favorable if std KELIM ≥ 1 or unfavorable if std KELIM < 1). In the low-risk group, no benefit from the addition of bevacizumab was found, regardless of std KELIM value. In the high-risk group, patients with favorable std KELIM had no survival benefit from bevacizumab (median OS within bevacizumab 48.2 months vs. 46.6 months, log-rank *p* = 0.7), whereas patients with unfavorable std KELIM derived the highest survival benefit from bevacizumab (median OS 29.7 vs. 20.6 months, log rank *p* = 0.1). With a log rank P of 0.1, the difference in survival was not statistically significant due to the small numbers of patients, but the difference was significant with the non-censored median survivals (Wilcoxon *p* = 0.004). 

Patients with high-risk disease and highly chemosensitive tumors may therefore not benefit from the addition of bevacizumab, while it may be useful in patients with poorly chemosensitive and high-risk disease. Due to the limited number of patients, the statistical power was reduced, and a validation in other datasets is warranted to confirm this hypothesis.

On the other hand, KELIM may also be useful in helping to select the patients who will have the highest benefit from PARP inhibitors. 

PARP inhibitors have recently changed the landscape of ovarian cancer treatment as a maintenance treatment option. Relationships between platinum-sensitivity and PARPi efficacy have been well established [50,51]. PARPi are more effective in patients with homologous recombination deficiency (HRD), especially in those with BRCA 1–2 mutations [52,53,54,55]. HRD is responsible for a defect in DNA double-strand break repair and is highly predictive of primary platinum sensitivity because tumor cells are unable to repair the double-strand breaks induced by platinum [56,57]. As a consequence, it was rational to assess the relationship between PARPi efficacy and KELIM as an indicator of the tumor primary chemosensitivity. 

A post hoc study from the data of 854 patients enrolled in VELIA trial explored the association between KELIM and the long-term clinical benefit of veliparib [42]. Analyses were performed according to the surgical groups of patients, comprising 700 patients treated with primary debulking surgery (PDS) and 154 patients treated with interval debulking surgery (IDS). The prognostic value of KELIM was confirmed since patients with favorable KELIM experienced higher PFS regardless of the treatment arm, in both PDS and IDS populations. In the PDS population, veliparib provided a significant PFS benefit in patients with favorable KELIM (HR 0.67, 95% CI 0.47–0.97), contrary to those with unfavorable KELIM (HR 0.77, 95% CI 0.56–1.06). In the IDS population, patients with favorable KELIM seemed to benefit the most from veliparib maintenance, but the difference was not significant due to the limited number of patients (median PFS, 29.8 months in the veliparib-throughout group and 20.8 months for the control group, HR 0.54, 95% CI 0.27–1.07). On the other hand, patients with unfavorable KELIM did not have benefit from veliparib (median PFS, 14.3 months in the veliparib-throughout group and 14.4 months in the control group, HR 0.87, 95% CI 0.41–1.87). Patients with higher tumor primary chemosensitivity, assessed by favorable KELIM, therefore seemed to benefit the most from the addition of veliparib. These results are consistent with those previously reported on the links between chemosensitivity and the benefit of PARP inhibitors. KELIM may be useful for identifying the patients who might benefit from veliparib in the cases where BRCA or HR status is not available.

### 5.4. An Online Calculator of KELIM

To simplify access to this tool, a calculator is available online that enables clinicians to rapidly calculate std KELIM during neoadjuvant (at http://www.biomarker-kinetics.org/CA-125-neo, accessed on 4 November 2021)) or adjuvant chemotherapy (at http://www.biomarker-kinetics.org/CA-125, accessed on 4 November 2021).

Clinicians are requested to enter the dates of chemotherapy cycles and the CA-125 values and dates during the first 100 days following the start of chemotherapy. The compute button enables calculation of KELIM and assessment of statistics regarding the risk of subsequent platinum-resistant relapse and of survival predictions (Figure 5). 

## 6. Conclusions

The backbone of the first-line treatment in advanced EOC patients is based on debulking surgery, meant to be complete with no visible residual lesion, and platinum-based chemotherapy followed by maintenance therapy [5]. 

While international guidelines have now widely recognized and incorporated the significant prognostic and therapeutic role of complete debulking surgery, the relevance of the tumor primary chemosensitivity has been insufficiently considered, although it may have a major impact on the success of the first-line medical–surgical treatment, on decision-making regarding the type of systemic therapy to prescribe, and on the comprehensive prognosis of patients [41].

The modeled CA-125 kinetic parameter KELIM, standing for ELIMination rate constant K, calculated during the first 100 days of neo-adjuvant or adjuvant chemotherapy has been established as a reproducible indicator of the tumor primary chemosensitivity on the data of more than 12,000 patients enrolled in seven large randomized clinical trials and a national cancer registry, along with a meta-analysis database (Table 1). 

All data converge for showing the strong prognostic impact of KELIM regarding the overall survival and the likelihood of cancer cure and also the potential utility for disease management in the first-line setting, especially regarding the surgery feasibility and utility or the selection of the best maintenance strategy. Additional data will be required to confirm this hypothesis. 

The prognostic value of KELIM will be prospectively assessed in the future randomized large phase II NIRVANA trial that will compare bevacizumab + niraparib vs. placebo + niraparib in patients operated with complete PDS. Moreover, being an indicator of patients who have particularly poor prognosis due to poorly chemosensitive disease that is not likely to benefit from PARP inhibitors, KELIM will be used to determine the best patients for innovative strategies meant to reverse chemoresistance, such as those with cell cycle checkpoint inhibitors or immunotherapy in a prospective SENSOVAR trial. 

## Figures and Tables

**Figure 1 cancers-14-00098-f001:**
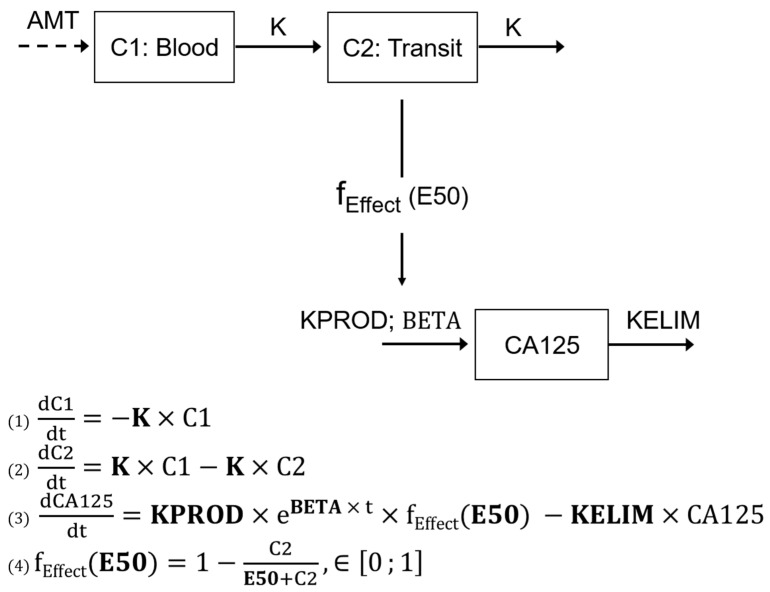
Description of the semi-mechanistic model. AMT: unknown dose amount; K: the treatment kinetic rate constant (days^−1^); KPROD: the zero order CA-125 tumor production rate (IU mL^−1^ days^−1^); BETA: tumor growth rate (days^−1^); KELIM: the first-order CA-125 elimination rate (days^−1^); f_EFFECT_: production inhibition; C1: central compartment receiving chemotherapy dosing; and C2: transit compartment to describe the treatment lag-time effect. CA0 corresponds to the estimated baseline CA-125 and “t” the time (Adjusted from You et al., Gynecol Oncol 2013 [21]).

**Figure 2 cancers-14-00098-f002:**
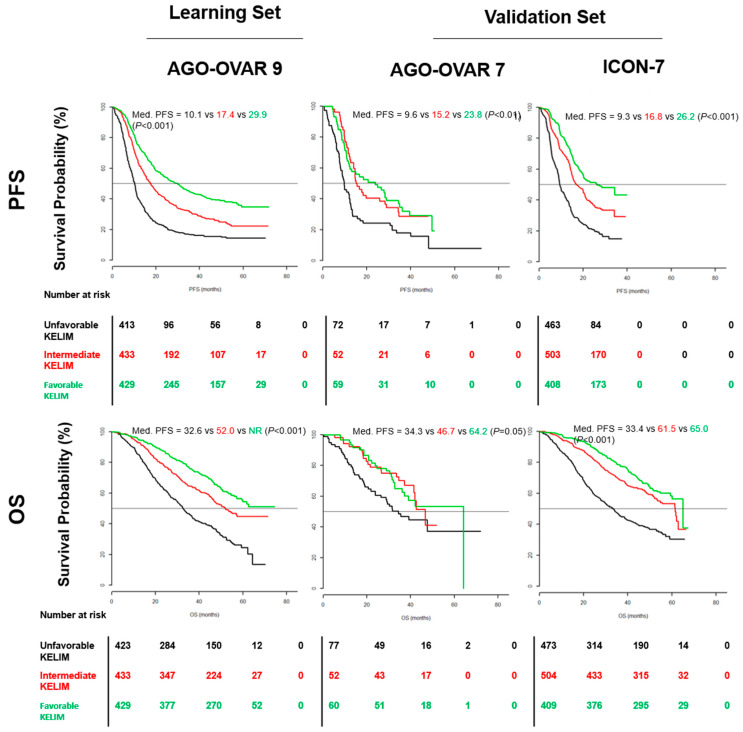
Kaplan–Meier analyses of PFS and OS in the learning set AGO OVAR 9 and the two validation sets AGO OVAR 7 and ICON-7. Black line, unfavorable KELIM tercile (minimum–0.05) days-1; red line, intermediate KELIM tercile (0.05−0.07) days^−1^; green line, favorable KELIM tercile (0.07–maximum) days^−1^; P: univariate log-rank tests; Med.: median. (Adjusted from Colomban et al., Clin Cancer Res 2019 [23]).

**Figure 3 cancers-14-00098-f003:**
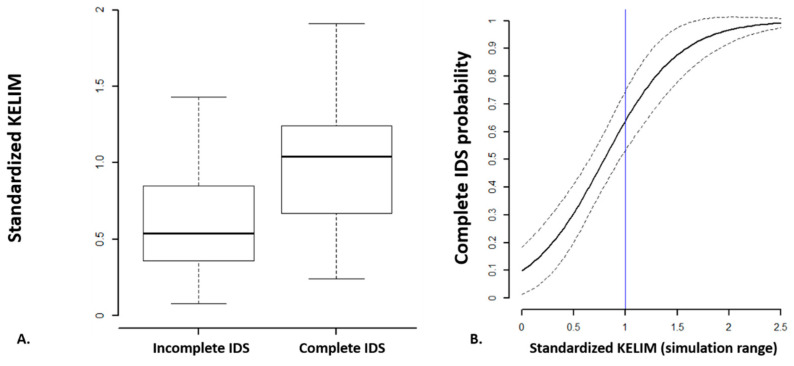
(**A**) Standardized KELIM distribution across surgery outcome. (**B**) Logistic regression curve regarding the probability of complete surgery according to continuous std KELIM values (Adjusted from You et al., Clin Cancer Res 2020 [27]).

**Figure 4 cancers-14-00098-f004:**
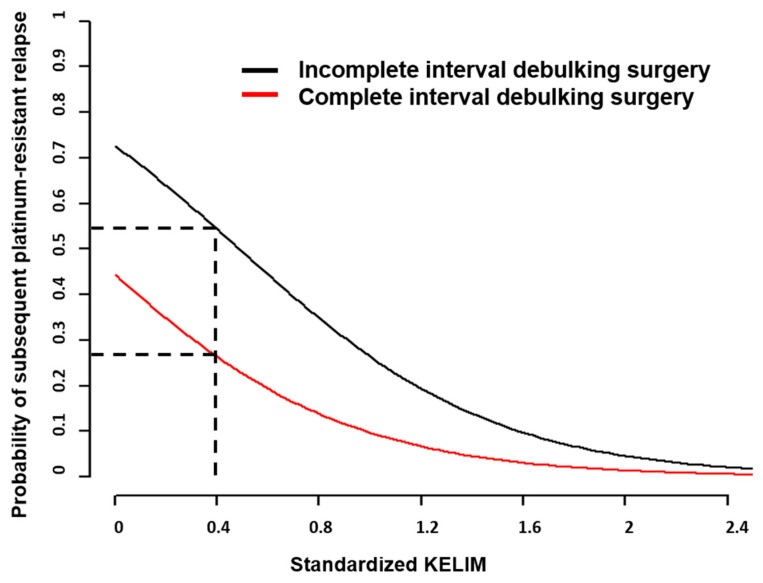
Platinum-Resistant Recurrence Score. Probability of subsequent platinum-resistant recurrence according to std KELIM. Dashed black line: illustration of a patient with std KELIM = 0.4; the risk of platinum-resistant relapse probability of 26% if IDS was complete or 54% if IDS was incomplete (Adjusted from You et al., Clin Cancer Res 2020 [27]).

**Figure 5 cancers-14-00098-f005:**
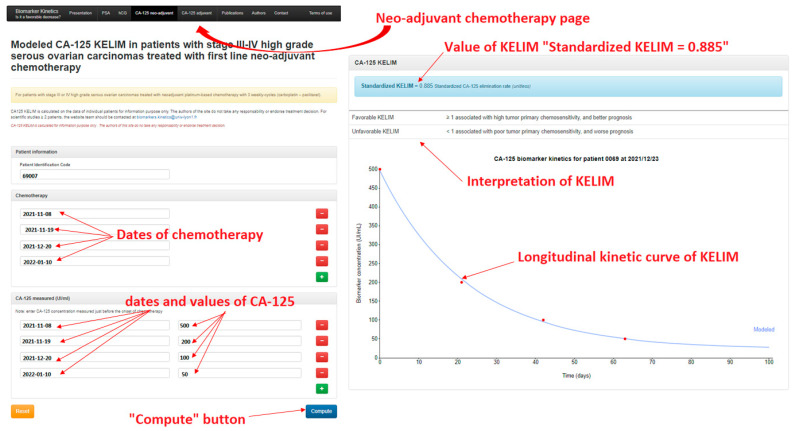
Illustration of the site http://www.biomarker-kinetics.org/CA-125-neo (accessed on 4 November 2021) for calculating KELIM in a patients treated with neo-adjuvant chemotherapy.

**Table 1 cancers-14-00098-t001:** Summary and main results of the articles on KELIM.

**Author**	**Situation**	**Population**	**Patients (n)**	**KELIM Prognostic Value in Multivariate Analyses**
You et al., 2013 [21]	ROC	CALYPSO	875	PFS (favorable vs. unfavorable) HR 0.53, 95% CI 0.45–0.61, *p* < 0.001
Colomban et al., 2019 [23,47]	Adjuvant	AGO-OVAR 7 AGO-OVAR 9 ICON 7	2868	PFS AGO-OVAR 7, unfavorable (reference), intermediate HR 0.41, 95% CI 0.24–0.68, favorable HR 0.47, 95% CI 0.30–0.76; AGO-OVAR 9, unfavorable (reference), intermediate HR 0.63, 95% CI 0.54–0.73, favorable HR 0.46, 95% CI 0.39–0.54; ICON 7, unfavorable (reference), intermediate HR 0.52, 95% CI 0.45-0.61, favorable HR 0.38, 95% CI 0.32–0.46 OS AGO-OVAR 7, unfavorable (reference), intermediate HR 0.53, 95% CI 0.31–0.92, favorable HR 0.58, 95% CI 0.35–0.97; AGO-OVAR 9, unfavorable (reference), intermediate HR 0.62, 95% CI 0.51–0.74, favorable HR 0.45, 95% CI 0.37–0.55; ICON 7, unfavorable (reference), intermediate, 0.51, 95% CI 0.42–0.61, favorable, 0.44, 95% CI 0.35–0.53 Risk of platinum-resistant relapse OR 0.17, 95% CI 0.11–0.25, *p* < 0.001
You et al., 2020 [27]	Neo adjuvant	CHIVA	134	PFS HR, unfavorable (reference); intermediate 0.50, 95% CI 0.31–0.79, *p* < 0.01; and favorable 0.36, 95% CI 0.21–0.62, *p* < 0.001 OS HR, unfavorable (reference); intermediate 0.31, 95% CI 0.17–0.55, *p* < 0.001; and favorable 0.28, 95% CI 0.16-0.50, *p* < 0.001 Probability of complete interval debulking surgery OR 16.13, 95% CI 5.51-53.38, *p* < 0.001 Risk of platinum-resistant relapse OR 0.13, 95% CI 0.03–0.49, *p* < 0.001
Wagensveld et al., 2020 *(Abstract)* [29]	Neo adjuvant	The Netherlands Cancer Registry	1582	PFS (favorable vs. unfavorable) HR 0.77, 95% CI 0.67–0.89, *p* < 0.001 OS (favorable vs. unfavorable) HR 0.75, 95% CI 0.65–0.87, *p* < 0.001 Probability of Complete interval debulking surgery OR 5.25, 95% CI 3.68–7.59, *p* < 0.001
You et al., 2020 *(Abstract)* [3]	Adjuvant Neo adjuvant	AGO-OVAR 7 AGO-OVAR 9 ICON 7 Netherlands Cancer Registry	4450	Probability of long progression-free survivorship OR 4.17, 95% CI 1.61–10.64
Corbaux et al. 2021 *(Abstract)* [31]	Adjuvant	GCIG mega-database	5842	PFS (favorable vs. unfavorable) HR 0.50, 95% CI 0.44–0.57 OS (favorable vs. unfavorable) HR 0.45, 95% CI 0.37–0.54
You et al., 2020 *(Abstract)* [42]	Adjuvant Neo adjuvant	VELIA	854	PFS (unfavorable vs. favorable) Immediate Primary Surgery Veliparib group: HR 0.61, 0.42–0.87; Control group: HR 0.69, 0.49–0.95 Delayed Primary Surgery Veliparib group: HR 0.56, 0.33–0.95; Control group: HR 0.64 0.39–1.06 Probability of complete delayed primary surgery 51.9% vs. 32.4%
				Association with Treatment Efficacy
You et al., 2021 *(Abstract)* [30]	Adjuvant Neo adjuvant	ICON8	1004	Weekly dose-dense chemotherapy vs. standard thrice-weekly chemotherapy PFS Immediate Primary Surgery Unfavorable KELIM: univariate HR 0.80, 95% CI 0.54–1.17, *p* = 0.25 Favorable KELIM: univariate HR 1.27, 95% CI 0.72–2.22, *p* = 0.40 Delayed Primary Surgery Unfavorable KELIM: univariate HR 0.84, 95% CI 0.62–1.13, *p* = 0.25 Favorable KELIM: univariate HR 1.14, 95% CI 0.86–1.51, *p* = 0.36 OS Immediate Primary Surgery Unfavorable KELIM univariate HR 0.75, 95% CI 0.50–1.14, *p* = 0.19 Favorable KELIM: univariate HR 1.05, 95% CI 0.53–2.06, *p* = 0.58 Delayed Primary Surgery Unfavorable KELIM univariate HR 0.80, 95% CI 0.60–1.08, *p* = 0.16 Favorable KELIM: univariate HR 0.90, 95% CI 0.65–1.24, *p* = 0.53
You et al., 2020 *(Abstract)* [42]	Adjuvant Neo adjuvant	VELIA	854	Benefit from veliparib Favorable KELIM: PFS benefit (HR 0.67, 95% CI 0.47–0.97 in PDS population; HR 0.54, 95% CI 0.27–1.07 in IDS population)
Colomban et al., 2020 [49]	Adjuvant	ICON 7	1386	Benefit from bevacizumab High risk group with unfavorable std KELIM: OS benefit (median 29.7 months, 95% CI 24.0–35.2 vs. 20.6 months, 95% CI 17.6–23.9, log-rank *p* = 0.1)

ROC = recurrent ovarian cancer; PFS = progression-free survival; OS = overall survival; *(Abstract)*: data from abstracts presented at congresses. The data may change at the final publication.

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
