# Peer review of "The Increasing Prognostic and Predictive Roles of the Tumor Primary Chemosensitivity Assessed by CA-125 Elimination Rate Constant K (KELIM) in Ovarian Cancer: A Narrative Review"

_cancers, 2021, doi:10.3390/cancers14010098_

Round 1
Reviewer 1 Report
The present review by Lauby et al. contextualizes available data on the "CA-125 ELIMination rate 3 constant K (KELIM)" in ovarian cancer and highlights potential utilities and addresses clinically highly debated topics such as prediction of IDS and adjuvant bevacizumab benefit.
The research group around You et al. thereby gives a sound and comprehensive summary of their own fruitful scientific work of the past 8 years since KELIM's first introduction for ovarian cancer in Gynecol Oncol 2013. The manuscript is clearly structured, study results are comprehensively discussed, and cited references are up to date.
Considering the constant increase of available clinical information, which may allow the attending physician (or the tumorboard) to weigh potential treatment options, the implementation of dynamic mathematical algorithms in clinical oncology is long overdue. In an era of personalized medicine, concepts like KELIM are likely to be the future, especially in cases like bevacizumab response prediction, for which no seemingly “simple” biomarkers are available (e.g. as CPS for CPI-therapy).
Even though the review comprises all relevant information of the astonishingly broadly validated KELIM, it however might miss the promising opportunity to reach a broad spectrum of gynecologic oncologists in its present form – and thereby achieve broad clinical implementation. Whereas KELIM’s clinical applicability and its broad validation are clearly described and the how-to of the online calculator in chapter 5.4 remarkably lowers the inhibition threshold to actually try out KELIM, the review completely lacks a understable summary what KELIM actually tries to depict (or more likely: to provide a surrogate for); the only relevant (and very brief) information is given in chapter 3.1., however, equally long discussed KPROD, BETA and E50 are obviously irrelevant for the main statement of the review.
If the review aims to actually reach non-statisticians, an easier description of the concept of KELIM is necessary. The easier its conception is communicated, the more likely it will be accepted among gynecooncologists. I may suggest adding a simple description of the concept after 3.1 with a simple figure, replacing or at least complementing Figure 1, which is a copy&paste of the 2013 Gynecol Oncol publication and does not provide clear and relevant information on KELIM. (If a personal note is allowed: As far I think I might have understood the statistical conception – in a simplified version: KELIM represents the slope in a time (t)-dependent function of the decay of CA-125 samples and thereby the -x in F(t)=exp(-x*t). If so, I may suggest providing an exemplary time-dependent CA125-course and mark KELIM as the slope, but the decision of how to depict it is at the discretion of the authors.)
Otherwise, I may congratulate the authors to their work and the comprehensive review.
Reviewer 2 Report
This is an excellent narrative review that summarizes and interpretes the current Knowledge on the Topic. It should , however be indicated in the title that it is a narrative review.
Reviewer 3 Report
In the present work the authors have reviewed the literature in the form of a narrative review, presenting an overview of KELIM utility across the different data published or presented in congress.
The review is well-written and the authors presented their case thoroughly. Their work has merit for publication after some minor corrections.
First of all, it is very important that such algorithmic approaches are of continuous and ongoing research. They are of outmost importance towards the prognosis of ovarian cancer and patient quality of life.
In the present review, it was kind difficult to follow the details of individual studies and therefore I would suggest to the authors to summarize their findings in table. They should include their results in such a manner that would be easy for the reader to find the information at one place.
I have made a small search in pubmed and found that there is not an enormous number of publications available on this topic, thus the authors could move to, although not imperative, a more systematic review on the subject.
In addition, I would like to have some information on the mathematics behind the algorithm. In particular, please mention how the rate constant is calculated (from the theoretical point of view).
Further on, (which is connected to my previous comment) it would be interesting for the reader, and especially for the non-expert, to have a small reference to the history of the algorithm.
Author Response
Please see the attachment

This manuscript is a resubmission of an earlier submission. The following is a list of the peer review reports and author responses from that submission.